# Low-Frequency Magnetic Resonance Imaging Identifies Hand Joint Subclinical Inflammation in Systemic Sclerosis

**DOI:** 10.3390/diagnostics12092165

**Published:** 2022-09-06

**Authors:** Bojana Stamenkovic, Sonja Stojanovic, Valentina Zivkovic, Dragan Djordjevic, Mila Bojanovic, Aleksandra Stankovic, Natasa Rancic, Nemanja Damjanov, Marco Matucci Cerinic

**Affiliations:** 1Institute for Treatment and Rehabilitation Niška Banja, 18205 Niška Banja, Serbia; 2Faculty of Medicine, University of Niš, 18000 Niš, Serbia; 3ENT Clinic, University Clinical Center Niš, 18000 Niš, Serbia; 4Institute of Public Health Niš, 18000 Niš, Serbia; 5Medical Faculty, University of Belgrade, 11000 Belgrade, Serbia; 6Department of Experimental and Clinical Medicine, University of Florence, 50121 Florence, Italy; 7Department of Geriatric Medicine, Division of Rheumatology and Scleroderma Unit, Azienda Ospedaliero Universitaria Careggi, 50134 Florence, Italy; 8Unit of Immunology, Rheumatology, Allergy and Rare Diseases (UnIRAR), IRCCS San Raffaele Hospital, 20132 Milan, Italy

**Keywords:** systemic sclerosis, magnetic resonance imaging, hand inflammation

## Abstract

Objectives: The aim of this work was to determine hand joint inflammation in systemic sclerosis (SSc); patients with rheumatoid arthritis (RA) with hand joint involvement were used as controls. Our investigation also aimed at examining the relationship between these subclinical inflammatory changes in the hands, verified by low-frequency MRI, and clinical (especially cardiopulmonary) manifestations, disease activity, and functional capacity in patients with diffuse cutaneous (dcSSc) and limited cutaneous SSc (lcSSc). Methods: Out of 250 SSc patients, the selection included 82 patients with signs and symptoms of joint involvement, and 35 consecutive RA patients. These patients underwent clinical and laboratory investigations, and hand X-ray and MRI of the dominant hand. Synovitis/tenosynovitis, bone edema, and erosions were investigated, and the bone changes were quantified and scored using the RAMRIS method. HAQ index, modified Rodnan skin score, examination of internal organ involvement, and serological markers for SSc, as well as rheumatoid factor (RF) and cyclic citrullinated peptides antibodies (ACPA), were performed on all experimental group subjects. Results: MRI of the dominant hand showed a significantly higher number of cases with synovitis (78%) than the number of patients with clinically swollen joints (17.1%; *p* < 0.001); bone edema was found in 62 (75.6%) SSc patients. MRI also showed a higher number of erosions (52; 63.4%) compared to those (22; 27.5%) detected with X-ray (*p* < 0.001). The average values of the total MRI score of synovitis/edema and erosions in the wrist (*p* < 0.001) and MCP joints (*p* < 0.001) were statistically higher in RA than in SSc patients (*p* < 0.001). The probability of the MRI-detected inflammatory changes was considerably higher in SSc patients who had vascular complications (digital ulceration, OR = 4.68; 95% IP: 1.002–22.25; *p* < 0.05), in patients with more severe functional impairment (OR = 8.22; 95% IP: 1.74–38.89; *p* < 0.01), and in patients with active disease (OR = 3.132; 95% IP: 1.027–9.551; *p* < 0.05). In our investigation, patients with a limited form of the disease and with inflammatory changes on MR more often had higher functional impairment compared to the other group without MRI inflammation. Conclusions: Our data show that in SSc MRI can detect a significant subclinical joint inflammation. RAMRIS confirmed the high degree of joint inflammation in RA, but also revealed a great deal of joint inflammation in SSc. That inflammation is associated with systemic inflammation (disease activity), vascular complications, and more severe forms of the disease, as synovitis cannot be precisely diagnosed by the clinical examination of joints. These results suggest that a careful joint investigation is necessary in SSc, and that in symptomatic patients, MRI may identify joint inflammation. In clinical practice, this evidence might drive to an early targeted therapy, thus preventing joint erosions.

## 1. Introduction

Systemic sclerosis (SSc) is a complex autoimmune disease characterized by vasculopathy, multiorgan fibrosis, and immune dysregulation [1].

In SSc, musculoskeletal manifestations (MSM) are frequently observed; in particular, arthralgias, arthritis, contractures, tendon friction rubs (TFRs), tenosynovitis, as well as myalgias, muscle tenderness, and myositis [2]. In this context, the assessment of arthritis may be challenging due to the presence of skin edema, thickening, or tethering, as well as the presence of digital ulcers [3], subcutaneous calcinosis [4], and atrophic contractures [2,5]. Earlier in the disease, clinical examination and laboratory and X-ray findings are neither sensitive nor specific enough to provide relevant information of joint inflammation. In practice, X-rays may detect late erosions as a consequence of synovitis, but cannot show the synovial inflammation or the early-stage degeneration of the cartilage and bones [5,6]. Today, we know that magnetic resonance imaging (MRI) is more sensitive and precise for assessing inflammatory joint changes to the hand when compared to clinical examination and X-ray findings in patients with rheumatoid arthritis (RA) [7]. A study on a small number of SSc patients has suggested the role of MRI in identifying and quantifying joint inflammation. Therefore, the use of MRI in SSc may be meaningful for the early detection of inflammatory changes in hand joints in order to provide early treatment. In fact, MRI may detect much earlier bone modifications that cannot be identified with X-rays, visualizing the inflammation in the synovial membrane, bursae, tendons, and ligaments. Furthermore, this method can be used for quantifying either cartilage loss and synovial membrane and the volume of synovial fluid, which are useful for monitoring the disease progression and therapeutic efficacy [7,8,9,10].

Our aim was to evaluate the presence of subclinical joint inflammation (synovitis, tenosynovitis, bone edema), as well as bone erosions and acro-osteolysis, on the dominant hand of symptomatic SSc patients. Furthermore, we wanted to establish the relationship between these subclinical inflammatory changes in the hands, verified by low-frequency MRI, and clinical disease activity (especially cardiopulmonary manifestations), as well as functional capacity, in patients with diffuse cutaneous (dcSSc) and limited cutaneous SSc (lSSc).

## 2. Patients and Methods

Out of 250 SSc patients followed up at the Rheumatology Clinic, Institute Niška Banja, Serbia, classified according to ACR/EULAR criteria [11], without clinical features of overlap syndromes or mixed connective tissue disease (MCTD), 82 patients (77 females, average age 54.4 ± 12.6, and disease duration 7.41 ± 6.12 years) complaining of hand arthralgia and/or tendon friction rub, swollen joints, and contractures were selected and divided into subsets [12] (59 limited cutaneous SSc and 23 diffuse cutaneous SSc). The patients positive for cyclic citrullinated peptides antibodies (ACPA) and rheumatoid factor (RF) were separated from the other negative SSc patients. A group of 35 RA patients (28 females, average age 54.28 ± 9.6, and average disease duration 6.8 ± 5.8 years) who met the 2010 RA ACR/EULAR criteria [13] were used as control. All patients provided a signed informed consent, and the study was approved by the Local Medical Ethical Committee of the Institute for Treatment and Rehabilitation, Niška Banja, and Faculty of Medicine, University of Niš, Serbia.

All SSc patients were scored with the Revised 2017 EUSTAR disease activity index (EScSG-AI criteria) [14] to detect systemic disease activity. Functional status was defined based on the Health Assessment Questionnaire (HAQ). Internal organ involvement was defined based on specific criteria [15], whereas muscle involvement was evaluated based on the presence of muscle tenderness and/or increased value of creatine phosphokinase (CPK). Skin involvement was measured with the modified Rodnan skin score [16], and the presence of digital ulcers and pitting scars was also recorded. Antinuclear (ANA) and anticentromere (ACA) antibodies were determined by means of indirect immunofluorescence on HEP-2 cells (Immunoconcepts, Sacramento, CA, USA). Titer 1:40, i.e., titer which was in practice recommended as “screening”, was taken as limit ANA titer (“cut off” titer, i.e., titer representing the lowest value accepted as positive test result). Anti-topoisomerase antibodies (ATA) were determined using the CIE method (Counter Immunoelectrophoresis), Imtec, Immunodiagnostics, Berlin, Germany. All patients were tested for the presence of rheumatoid factor (RF) for values > 20 IU/L, which were considered positive. The ELISA (enzyme-linked immunosorbent assay) method, Imtec, was used to determine the presence of anti-CCP antibodies, considered positive when exceeding > 25 U/mL. All RA patients were scored for disease activity with the DAS28SE.

Standard hand and wrist X-ray was performed in all patients with SSc and RA. The changes were evaluated by a radiologist and rheumatologist blinded to clinical and serological patient data. The following features were investigated: focal or diffuse joint space narrowing, erosions (cortical bony surface discontinuity), juxta-articular or generalized osteoporosis and acro-osteolysis, tissue calcifications, and flexion contractures (in these cases, joint space narrowing at proximal interphalangeal (PIP) joints and distal interphalangeal (DIP) joints were not analyzed).

In SSc and RA patients, MRI joint visualization was conducted with a 0.2T Artroscan MRI Unit (Esaote Biomedica, Genoa, Italy) with special surface coil. Aside from an axial and coronal view of standard T1W and T2W sequences, a specially created fat-suppression sequence, STIR (Short Time Inversion Recovery), was used. The stated MRI sequences on dominant hand [17] were used to scan metacarpophalangeal (MCP) joints, i.e., from the second to the fifth, as well as the metacarpal base of all carpal bones, distal radius, and ulna. A contrast agent (gadolinium) IV bolus injection 0.1 mmol/kg (Gadopentate Dimeglumine-Magnevist^®^, Gd DTPA, Schering) was used for differentiating the intensity of the synovial signal. MRI visualization results were analyzed by the OMERACT (Outcome Measures in Rheumatology) RAMRIS (Rheumatoid Arthritis Magnetic Resonance Imaging Scoring System) method [7,18]. Two radiologists, blinded to disease diagnosis, interpreted the results, and gave their professional opinion on the presence of synovitis, tenosynovitis, and bone lesions.

### 2.1. Inflammation Assessment Using MRI

The assessment of inflammation with the MRI scan allowed the scoring of synovitis, bone lesions (bone erosions and bone edema), and the presence of tenosynovitis (Appendix A).

#### 2.1.1. Synovitis Score

Synovitis detected on MRI was scored after contrast-enhanced signal intensity in the synovial tissue at T1W SE (spin echo) and STIR sequence: the score range was from 0 to 3, with 0 being normal synovial tissue, whereas score 1, 2, and 3 indicated mild, moderate, and severe synovial thickening (thicker than joint capsule), respectively. The total score was monitored for each of the 4 MCP joints (excluding MCP 1), the total sum being from 0 to 12.

The total score for the wrist was monitored in three areas: radiocarpal, radioulnar, and carpometacarpal joints (range 0–9).

#### 2.1.2. Bone Lesion Score

Bone erosions were assessed on a two-plane MRI scan. Erosions implied cortical bone interruptions at minimum on one plane, and were scored from 0 to 10 depending on the size and volume of trabecular bone loss. Erosion assessment was also conducted on the wrist bones and MCP joints, i.e., from the second to the fifth joint.

Bone edema was assessed either as an isolated condition or as a condition defined by gradually blurring edges that surrounded the erosion with a high signal intensity at the T2W STIR MRI sequence.

Bone erosion and edema were assessed at 15 locations (distal end of the radius; the head of the ulna; wrist bones: scaphoid bone, lunate bone, triquetral bone, pisiform bone, trapezium, trapezoid bone, capitate bone, hamate bone; at bases; from the first to the fifth metacarpal bone; and at eight locations (metacarpal heads and phalanx bases from the second to the fifth MCP joint). The total score of carpal bone erosions ranged from 0 to 150, whereas the score for MCP joints was 0–80. The total edema score was 0–24 for MCP joints and 0–45 for carpal bones.

#### 2.1.3. Assessment of Tenosynovitis

The MRI scan was evaluated based on the presence of a high-intensity signal at the T2W STIR sequence followed by appropriate localization. After administration of the gadolinium-based contrast agent, tenosynovitis was confirmed in T1W SE sequences: 5 tendon groups (extensor pollicis longus and brevis, extensor carpi radialis longus and brevis, extensor digitorum, extensor digiti minimi, and extensor carpi ulnaris) in the wrist, i.e., analyzed at dorsal side of the joint; and 2 tendon groups (flexor carpi radialis, flexor digitorum profundus, and superficialis) analyzed at the palmar side. The extensor and flexor tendon groups in MCP joints, i.e., from the first to the fifth MCP joint at the dorsal and palmar side of the joint, were assessed.

### 2.2. Statistical Analysis

Statistical parameters used for data overview included arithmetic means ± standard deviation (S.D.), medians, and index structure (%). The average values of continuous variables in the two tested groups were compared by means of Student’s *t*-test or Mann–Whitney test (depending on data distribution normality for continuous variables). Fisher’s exact test was used for comparing the frequency of category features. Mutual dependence between the values of different characteristics was assessed by applying Pearson’s correlation coefficient (r). Student’s *t*-test was used for testing the statistical significance of correlation coefficients. The assessment of the association between the examined factors and the probability of the detection of inflammatory changes, synovitis, bone edema, erosions, and tenosynovitis in the subjects of the experimental group was performed by logistic regression analysis. The approximate values of relative risk were calculated (odds ratio—OR), as well as their 95% confidence interval. The estimation of the statistical significance of the OR value was performed by calculating the Wald value. Significance was set at *p* < 0.05.

The results of statistical analysis were shown in tables and graphs. Calculations were performed using SPSS software, V 23.0.

## 3. Results

According to the EUSTAR disease activity index, 47/82 (57.3%) SSc patients had an active disease, 12 had a positive RF (14.6%), and 11 had a positive ACPA. The characteristics of SSc patients are shown in Table 1.

Out of 82 SSc patients, swollen joints were detected in 14 (17.1%), arthralgia in 66 (80.5%), tendon friction rubs in 12 (14.6%), and joint contractures in 28 (34.1%) patients.

A positive correlation was found between ACPA-positive patients and swollen joints (*p* < 0.0001) and also bone erosions (*p* < 0.0002).

Out of 35 RA patients, 26 (74.3%) were RF-positive, whereas 27 (77.1%) were ACPA-positive. In the same group, 20 (57.1%) patients had C-reactive protein (CRP) > 6 mg/dL. The average DAS 28-SE was 5.2 ± 1.34 (range: 2.1 to 7.2).

Overall, the clinical and MRI findings were compatible in 14 (17.1%) patients with positive and in 18 (22.0%) patients with negative findings (total of 32 (39.1%) patients), whereas 50 (61.0%) patients had a clearly positive MRI, but negative clinical findings (Table 2).

### Comparing MRI Scan Results with Clinical and Radiographic Musculoskeletal Manifestations in Patients with Systemic Sclerosis

Among 82 SSc patients, MRI showed the presence of synovitis in a very high number of patients (78.0%), which was much more significant than what was observed with the clinical analysis (17.1%; chi-squared test: *p* < 0.001). On the other hand, tendon friction rubs, clinically detected in 14.6% of patients, were confirmed by MRI in 14.1% of patients (*p* = 0.375). The number of clinically detected joint contractures (34.1%) was very close to that seen at X-ray (32.1%, *p* = 0.617). On X-rays, the erosions were observed in 27.5% of patients, whereas with MRI, they were more frequently detected (in 63.4% patients, *p* < 0.001). Similarly, on the wrist (51.2 vs. 11.5%, respectively; *p* < 0.001) and on MCP joints (45.1 vs. 9.1%, respectively, *p* < 0.001), MRI was able to more frequently detect erosions than X-ray.

On X-ray, acro-osteolysis was found in 17 (21.3%) patients, and with MRI, in 22 (27.5%) patients (*p* = 0.357) (Table 3).

No statistical difference was found between the total MRI score of synovitis, edema, and erosions in the wrist and MCP joints of the dominant hand between lcSSc and dcSSc (Appendix A).

The average value of the total MRI score of synovitis, edema, and erosions in the wrist of the dominant hand was statistically more significant in RA than in SSc patients (*p* < 0.001). Similarly, the average values of the total MRI score of synovitis, edema, and erosions in MCP joints of the dominant hand were statistically much higher in RA than in SSc patients (*p* < 0.001) (Table 4).

In the wrist, the localization of the highest total score of synovitis, either in SSc or RA, was detected in the distal radio-ulnar joints (DRUJ), and as expected, edema was significantly higher in RA than in SSc patients (*p* < 0.001). In SSc, the highest total score of erosions and edema was localized in the capitate bone area, whereas in RA patients, it was localized in the lunate bone, still statistically more numerous than in SSc patients (*p* < 0.001). In MCP2/3/5 joints, the highest total score of synovitis (*p* < 0.01) and erosion score were higher in RA patients (*p* < 0.001) (Table 5).

The number of patients with synovitis (100.0 vs. 63.4%), erosions (94.3 vs. 51.2%), and bone edema (100.0 vs. 61%) in the dominant hand wrist was significantly higher in RA (*p* < 0.001), as well as the number of patients with inflammatory changes (synovitis 100.0 vs. 58.5%: erosion 91.4 vs. 45.1%; bone edema 97.1 vs. 56.1%, at MCP joints (*p* < 0.001)).

In SSc patients, MRI showed erosions in 52 (63.4%) patients, whereas X-ray showed erosions in 22 (27.5%) patients only (chi-squared test: *p* < 0.001): 18 (22.5%) patients had both positive X-ray and MRI findings, whereas 26 (32.5%) patients had both of these findings as negative. In four patients (5.1%), erosions were detected by X-ray only, whereas in 32 (40.0%) patients, erosions were found only by MRI. In RA, MRI confirmed erosions in 34 (97.1%) patients, whereas upon X-ray, erosions were found in six (17.1%) patients only (chi-squared test: *p* < 0.001) (Table 6).

Out of 65 patients with SSc in whom inflammation was detected by MRI, univariate logistic regression analysis showed digital ulcerations in 25 (38.5%) cases, which was a significantly higher frequency compared to that in patients without inflammation (2 out of 17 patients; 11.8%; OR = 4.68; 95% IP: 1.002–22.25; *p* < 0.05). Patients with inflammation were significantly more likely to have HAQ values greater than 1.5 (52.3 vs. 11.8%; OR = 8.22; 95% IP: 1.74–38.89; *p* < 0.01), but also an active disease (OR = 3.132; 95% IP: 1.027–9.551; *p* < 0.05), as compared to the group without inflammation (Table 7).

Patients with acro-osteolysis had a statistically significant higher erosion score compared to those without acro-osteolysis (20.68 ± 34.17 vs. 6.85 ± 10.24, *p* < 0.05), and also had a statistically significant higher disease activity (5.23 ± 1.74 vs. 3.43 ± 1.77 *p* < 0.0001) (Appendix A).

Out of the 46 patients with lSSc in whom inflammation was verified by MRI, 21 (45.7%) patients had HAQ values higher than 1.5, which was significantly more often than in patients without inflammation (1 out of 13 patients; 7.7%; OR = 10.08; 95% IP: 1.20–84.05; *p* < 0.05). The differences in values and frequency of all examined characteristics in subjects with dSSc with and without MRI-detected inflammation were not statistically significant. The results were obtained by multivariate logistic regression analysis (Table 8).

A significant positive correlation between synovitis score (r = 0.371, *p* < 0.001), erosion score (r = 0.229, *p* < 0.001), and edema score (r = 0.279, *p* = 0.016) was found on MRI and EScSG-AI (Figure 1) in all SSc patients. However, no significant correlation between RAMRIS score and Rodnan skin score, disease duration, disease subset, and antibodies was found in SSc.

In RA, a significant correlation was confirmed between synovitis score, erosion score (r = 0.698 and *p* < 0.001), and edema (r = 0.622 and *p* < 0.001) on MRI and disease activity score, DAS 28SE (r = 0.344 and *p* < 0.05).

## 4. Discussion

Our data show that in SSc, inflammatory joint involvement is a rather frequent subclinical manifestation in symptomatic patients [19,20,21,22]. In our patients, the clinical and X-ray evaluation was not sensitive enough to identify the whole range of inflammatory changes either at the articular or periarticular level. In the literature, MRI was found to accurately detect soft tissue inflammation, proving it to be more sensitive than ultrasonography for identifying inflammation in patients with arthralgia without evident signs of synovitis [23,24]. In our patients, the contrast (gadolinium) provided significant visualization of inflamed synovial membrane, and it also showed evidence of the edema of the bone and soft tissue structures.

For assessing erosions in extremities, low-frequency MRI is considered equivalent to high-frequency MRI, but it is less sensitive in assessing bone edema, and requires the use of a contrast agent for the adequate detection of synovitis [7,17]. Unlike other methods, which include full-body scan, low-frequency MRI has numerous advantages, such as flexible patient positioning, affordable price, and reduced claustrophobia. The disadvantages of this method are reduced spatial resolution and a limited number of possible imaging techniques compared to full-body scan [25,26,27].

Our aim was to determine the capacity of low-frequency MRI to identify inflammatory hand changes in SSc patients, correlating the results with clinical and X-ray findings, and comparing the results with those obtained in RA patients. Furthermore, we wanted to establish the relationship between these subclinical inflammatory changes and clinical disease activity, as well as the functional capacity in patients with dcSSc and lSSc. 

The results of this study have shown a surprisingly significant number of inflammatory changes in the hands of SSc patients. Based on clinical examination, synovitis was suspected in 14/82 patients (17.1%) patients, but MRI was able to detect synovitis in 64/82 (78%) patients that were apparently clinically negative despite being symptomatic. The fact that 61% of our patients with positive MRI findings did not have positive clinical findings of synovitis definitively stresses the important role of MRI for diagnosing synovitis in SSc. Moreover, MRI could detect erosions in a significant number of patients (52/82 patients (63.4%)), compared to 22/82 (27.5%) detected by X-ray. Moreover, 40% of SSc-positive patients detected by MRI did not have any positive X-ray finding.

It is clear that MRI is an added imaging tool which may show inflammatory joint modifications when the clinical and X-ray findings are still negative. In practice, this evidence may help to design a targeted disease-modifying therapy to prevent the progression of joint damage and functional impairment in SSc patients as well [28,29,30,31,32,33].

Previously, the presence of MRI-detected erosive arthritis was confirmed in 41% of SSc patients [34], showing that MRI was more sensitive in assessing erosions than X-ray. The authors concluded that it was not clear if the erosive changes could be related to the overlap between SSc and RA or if they were merely the result of inflammatory changes in SSc. However, the selected group consisted of patients with clinical findings of arthritis, accompanied by serology which pointed to RA (RF+, ACPA+) [35,36]. Allanore et al. [37] examined hand vascular involvement by MRI angiography in 38 SSc patients, and confirmed that erosions were present in 16% of patients, whereas synovitis and tenosynovitis were recorded in 50% and 11% of patients, respectively. The possible explanation of the lower frequency of erosions in the French study could be due to the shorter duration of the disease in the SSc group (in 42% of cases, the disease lasted less than 5 years). Using a high-frequency MRI, Akbayrak et al. [25] found a significantly higher percentage of erosions (87%), similar to Chitale et al. (75%) [23] and to our study (63.4%).

Acro-osteolysis is the most common bone disorder in SSc, which occurs in 20–25% of patients [38]. Acro-osteolysis may be diagnosed by physical examination and X-ray (the gold standard for the detection), and recent studies have confirmed a comparable sensitivity of X-ray and ultrasonography (US) [39,40]. Our study showed a similar sensitivity of X-ray (17, 21.3%) and MRI (22, 27.5%) in detecting acro-osteolysis. Research on the prevalence of acro-osteolysis and its association with vascular changes and other systemic manifestations pointed out the link with a severe systemic disease, pulmonary arterial hypertension, digital ulcers, and late capillaroscopic pattern [41,42,43]. In agreement with the literature, our data confirm that SSc patients with acro-osteolysis had a significantly higher disease activity.

In our patients, MRI was significantly more sensitive for diagnosing synovitis, joint effusion, and tenosynovitis than clinical examination. Previously, low-frequency MRI of dominant hand has shown synovitis in all eight (100%) investigated SSc patients, whereas tenosynovitis was confirmed in 88% of patients [23]. An extremely high percentage of synovitis (87.5%), tenosynovitis (81.3%), and erosions (62.5%) was also detected [24], which is almost identical to our study. In our patients, MRI identified synovitis in 78% of the cases, and this result is close to that obtained by Akbayrak al (68%) [25]. The frequency of tenosynovitis seen in our study (14%) was similar to previous results (11%) [37].

It is well known that MRI is the only tool for detecting bone edema, which, as proved in RA, represents a pre-erosive change characterized by osteitis (cellular inflammatory infiltrate in subchondral bone, which can be a precursor of pre-erosive lesions) [44,45]. In our cases, we could quantify inflammatory changes and compare the degree of hand inflammation in SSc vs. RA patients with the RAMRIS, thus scoring the MRI-detected inflammatory changes in wrists and MCP joints (bone edema score, synovitis score, and erosion score) [46,47]. In our RA patients, bone edema occurred more frequently (*p* < 0.001) and had a higher statistically significant score (*p* < 0.001) in wrists and MCP joints than in SSc patients. Moreover, bone edema (seen 75.6% of patients) correlated with CRP in SSc patients, which might also be regarded as a marker of joint involvement in SSc. It still remains to be determined whether bone edema could be a marker pointing to a more aggressive disease requiring prompt treatment also, because MRI detected a significant percentage of synovitis in those patients that were clinically/radiologically negative. This evidence strongly suggests that MRI might be promptly used as a screening tool for subclinical hand inflammation in symptomatic patients only, in order to achieve an early diagnosis and treatment.

The localization of inflammatory changes found in the wrist and MCP joints of SSc patients is in agreement with previous studies [48]. Low et al. [34] confirmed the most frequent localization of synovitis in the radioulnar joint in SSc patients, whereas the study by Akbayrak et al. [25] showed the presence of synovitis in intercarpal joints. In our SSc patients, we recorded the highest percentage of erosions and bone edema in the capitate bone, whereas Akbayrak et al. [25] documented the highest percentage in the lunate bone.

Our research has shown that the probability of the MRI-detected inflammatory changes was considerably higher (over four times) in SSc patients who had vascular complications (digital ulcerations), in patients with more severe functional impairment (HAQ values > 1.5), and in patients with active disease, which was one of the most important findings. In our study, patients with a limited form of the disease, with inflammatory changes on MR, more often had higher functional impairment compared to the other group without MR inflammation.

The data from EUSTAR Database (2010) also show that synovitis, joint contracture, and tendon friction rubs are associated with a more severe disease and with systemic inflammation [20].

Concerning the prevalence of synovial involvement in SSc determined by power Doppler ultra-sonography, a recent study has shown that 32% of 103 patients had ultrasonographic tenosynovial involvement. Inflammatory synovitis was more frequent in the wrist and MCP joints, which is in line with our results. Sclerosing tenosynovitis (TS) was more frequent in men, and was associated with anti-RNA polymerase III antibodies, diffuse SSc, interstitial lung disease, and inflammatory arthralgia [49]. The data from an interesting, prospective, cross-sectional study, using grayscale and Doppler musculoskeletal ultrasound, show that the Scleroderma Health Assessment Questionnaire–Disability Index correlated with musculoskeletal ultrasound erosions. The skin score correlated with tendinitis grayscale [50].

In a prospective follow-up of 9165 EUSTAR patients, using univariate analysis, synovitis and tendon friction were identified as precursors of the progression of skin changes. Multivariate analysis showed that, with regard to disease subsets and the status of antibodies, synovitis and tendon friction were also the precursors of the progression of skin changes and cardiovascular manifestations, as well as the occurrence of ischemic digital ulcers, left chamber dysfunction, and renal crisis [51].

Our study has a number of limitations. Only symptomatic patients were included in the study. We recommend that further studies involve a group of patients without joint symptoms. The major limitation is that, being a cross-sectional study, it included patients at various stages of the disease. There is no possibility for the prospective monitoring of disease activity or therapeutic effect, which should be our goal in the coming period. The lack of a control group of healthy people is another limitation. A comparison with a control group of healthy people would be more adequate and would give more effective conclusions.

## 5. Conclusions

Our data show that there is a high subclinical joint inflammation in SSc, and that MRI was more sensitive than clinical and radiographic examinations in detecting the severity and degree of synovitis, bone edema, erosions, and tenosynovitis.

The importance of our research for the detection of MRI-verified inflammation of the dominant hand joints in SSc patients lies in the finding that inflammation is associated with systemic inflammation (disease activity), vascular complications, and more severe forms of the disease accompanied by interstitial lung disease, as synovitis cannot be precisely diagnosed by clinical examination of the joints. A significant correlation between MRI-detected synovitis, bone edema and erosion scores, and disease activity in SSc may point to the mechanisms of inflammatory changes in joints and their mutual dependence in terms of joint involvement and functional disability. The detection of the subclinical inflammation of hands in SSc using MRI scan has highlighted the need to apply this method in patients with specific symptomatology, all with the aim of initiating early contemporary treatment.

In conclusion, the problem of joint involvement in SSc should be more frequently addressed, and MRI should be employed as a standard of care in symptomatic patients to guide the therapy to target this neglected manifestation of the disease.

## Figures and Tables

**Figure 1 diagnostics-12-02165-f001:**
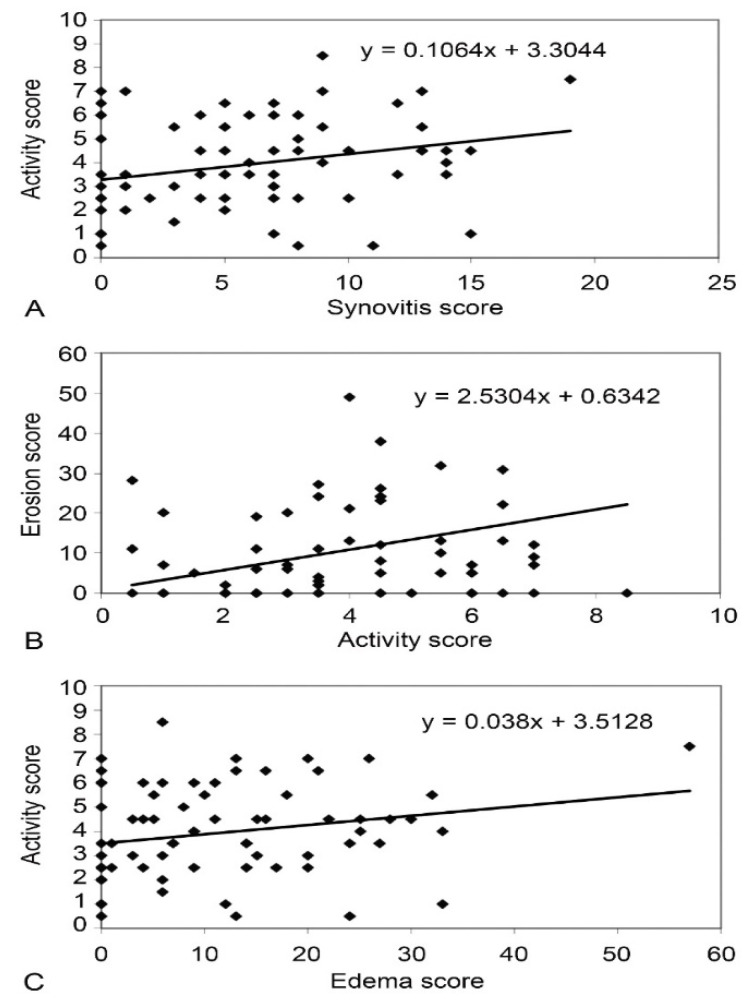
Correlation between synovitis score and EScSG-AI in patients with SSc (**A**), between EScSG-AI and erosion score in SSc group (**B**), and between edema score and EScSG-AI in patients with SSc (**C**).

**Table 1 diagnostics-12-02165-t001:** Characteristics of patients with systemic sclerosis (SSc).

Characteristics	Patients with SSc (*n* = 82)
Age	54.45 ± 12.51
Men/women	5 (6.1%)/77 (93.9%)
Duration of disease (years)	8.41 ± 6.12
Cutaneous subtype	
Limited	59 (72.0%)
Diffuse	23 (28.0%)
Raynaud’s phenomenon	80 (97.6%)
Digital ulcerations	27 (32.9%)
HAQ	1.22 ± 0.74
HAQ > 1.5	36 (43.9%)
Muscle weakness	5 (6.1%)
Lung fibrosis	42 (51.2%)
Increased systolic pulmonary arterial pressure	17 (20.7%)
Kidney involvement	8 (9.8%)
Positive antinuclear antibodies	63 (77.8%)
Positive antitopoisomerase-1 antibody	14 (17.1%)
Positive anticentromeric antibodies	23 (29.5%)
Elevated creatine phosphokinase	3 (3.7%)
Increase in acute phase reactants	30 (36.6%)
Positive RF	12 (14.6%)
Positive anti-CCP At	11 (13.4%)
Proteinuria	-
Active disease according to the European score (EScSG-AI)	47 (57.3%)

**Table 2 diagnostics-12-02165-t002:** Prevalence of synovitis in clinical findings compared to MRI findings.

Findings		MRI Synovitis		Total
		No	Yes	
Clinical Swollen Joints	No	18 (22.0%)	50 (61.0%)	68 (82.9%)
	Yes	-	14 (17.1%)	14 (17.1%)
	Total	18 (22.0%)	64 (78.0%)	82 (100.0%)

Data are *n* (%) unless stated otherwise.

**Table 3 diagnostics-12-02165-t003:** Musculoskeletal manifestations in systemic sclerosis assessed by clinical findings, radiographic findings, and magnetic resonance imaging (MRI).

Joint Manifestations	Clinical Finding (*n* = 82)	Radiographic Finding (*n* = 82)	MRI (*n* = 82)	*p*
**Joint Involvement**				
Synovitis	14 (17.1%)		64 (78.0%)	<0.001
Arthralgia	66 (80.5%)			
Tendon friction rubs	12 (14.6%)		11 (14.1%)	0.375
Joint contracture	28 (34.1%)	25 (32.1%)		0.617
Erosions		22 (27.5%)	52 (63.4%)	<0.001
Wrist		9 (11.5%)	42 (51.2%)	<0.001
MCP		7 (9.0%)	37 (45.1%)	<0.001
PIP		10 (12.8%)		
DIP		16 (20.5%)		
Joint space narrowing		36 (46.2%)		
Wrist		20 (25.6%)		
MCP		27 (34.6%)		
PIP		29 (37.2%)		
DIP		28 (35.9%)		
RÖ arthritis		14 (17.9%)		
Wrist		19 (54.3%)		
MCP		15 (42.9%)		
PIP		6 (17.6%)		
**Bone Tissue Involvement**				
Radiological demineralization		24 (29.6%)		
Acroosteolysis		17 (21.3%)	22 (27.5%)	0.357
Bone edema			62 (75.6%)	
Soft tissue involvement				
Flexion contractures	28 (34.1%)	25 (32.1%)		0.617
Calcinosis		31 (38.8%)		

Data are presented as *n* (%) unless stated otherwise. MCP: metacarpophalangeal; PIP: proximal interphalangeal; DIP: distal interphalangeal.

**Table 4 diagnostics-12-02165-t004:** Average values of total MRI score of synovitis, edema, and erosions in the wrist and MCP joints of dominant hand in SSc and RA.

Localization (Min. and Max. Values)	MRI Score in SSc	MRI Score in RA	*p*
**Wrist**			
Synovitis (0–7)	2.69 ± 2.29	4.37 ± 1.31	<0.001
Erosions (0–73)	6.58 ± 10.89	20.57 ± 10.23	<0.001
bone edema (0–33)	6.84 ± 7.43	18.60 ± 5.01	<0.001
**MCP Joints**			
Synovitis (0–12)	3.15 ± 2.95	5.26 ± 2.09	<0.001
Erosions (0–80)	3.99 ± 9.82	10.51 ± 7.90	<0.001
bone edema (0–24)	4.04 ± 4.76	9.69 ± 4.27	<0.001

Data are presented as mean ± S.D. unless stated otherwise. SSc: systemic sclerosis; RA: rheumatoid arthritis; MCP: metacarpophalangeal.

**Table 5 diagnostics-12-02165-t005:** Localization of the highest total MRI score in the wrist and MCP joints of dominant hand in patients with SSc and RA.

Localization	Patients with SSc	Patients with RA
**Wrist**		
synovitis	DRUJ (50; 61.0%)	DRUJ (35; 100%)
erosions	capitate bone (33; 40.2%)	lunate bone (32; 91.4%)
bone edema	capitate bone (46; 56.1%)	lunate bone (35; 100%)
**MCP Joints**		
synovitis	MCP2 (46; 56.1%)	MCP2 (34; 97.1%)
erosions	MCP2 (28; 34.1%)	MCP3 (32; 91.4%)
bone edema	MCP3 (38; 46.3%)	MCP3, MCP5 (34; 97.1%)

Data are presented as *n* (%) unless stated otherwise. SSc: systemic sclerosis; RA: rheumatoid arthritis; MCP: metacarpophalangeal; DRUJ: distal radio-ulnar joints.

**Table 6 diagnostics-12-02165-t006:** Prevalence of erosions in MRI compared to radiography in SSc and control RA groups.

Erosions—MRI		Erosions—Radiography		Total
		No	Yes	
RA group	No	1 (2.9%)	-	1 (2.9%)
	Yes	28 (80.0%)	6 (17.1%)	34 (97.1%)
	Total	29 (82.9%)	6 (17.1%)	35 (100.0%)
SSc group	No	26 (32.5%)	4 (5.1%)	30 (37.5%)
	Yes	32 (40.0%)	18 (22.5%)	52 (63.4%)
	Total	58 (72.5%)	22 (27.5%)	82 (100.0%)

Data are presented as *n* (%) unless stated otherwise. SSc: systemic sclerosis; RA: rheumatoid arthritis.

**Table 7 diagnostics-12-02165-t007:** Assessment of the association between the examined factors and probability of MRI-detected inflammatory changes in subjects with SSc; results of univariate logistic regression analysis.

Characteristics	OR	95% IP Limits for OR		*p*
		Lower	Upper	
Age	1.024	0.982	1.069	0.271
Duration of disease	1.061	0.963	1.168	0.233
Female sex	0.953	0.100	9.128	0.967
Duration of disease longer than 5 years	1.143	0.372	3.511	0.816
Diffuse SSc	0.745	0.215	2.578	0.642
Raynaud’s phenomenon	0.000	0.000	.	0.999
Digital ulcerations	4.687	1.002	22.256	0.049
HAQ > 1.5	8.226	1.740	38.896	0.008
Resorption of the distal phalanx of the Fingers	3.488	0.727	16.732	0.118
Muscle weakness	1.049	0.110	10.048	0.967
DLCO < 75%	1.971	0.566	6.867	0.286
FVC < 75%	1.130	0.277	4.616	0.864
sPAP > 30 mmHg	5.224	0.641	42.563	0.122
Positive ANA	1.099	0.309	3.904	0.884
Positive anti topo-1 At	4.000	0.485	32.984	0.198
Positive ACA	0.447	0.143	1.398	0.166
Elevated creatine phosphokinase		0.000		0.999
Positive CRP	2.656	0.549	12.841	0.224
Positive RF		0.000		0.999
Active disease	3.132	1.027	9.551	0.045

**Table 8 diagnostics-12-02165-t008:** Characteristics of the disease in SSC patients depending on the presence of MRI-detected inflammation.

Characteristics	SSc Patients (*n* = 82)				Patients with lSSc (*n* = 59)				Patients with dSSc (*n* = 23)			
	With Inflammation (*n* = 65)	Without Inflammation (*n* = 17)	*p*	OR (95% CI)	With Inflammation (*n* = 46)	Without Inflammation (*n* = 13)	*p*	OR (95% CI)	With Inflammation (*n* = 19)	Without Inflammation (*n* = 4)	*p*	OR (95% CI)
Age	55.23 ± 13.18	51.47 ± 9.27	0.271	1.02 (0.98–1.06)	56.39 ± 11.77	52.46 ± 7.70	0.260	1.03 (0.97–1.09)	52.42 ± 16.10	48.25 ± 14.24	0.621	1.01 (0.94–1.09)
Duration of disease	8.82 ± 6.40	6.82 ± 4.70	0.233	1.06 (0.96–1.16)	9.10 ± 6.68	6.77 ± 5.03	0.249	1.06 (0.95–1.18)	8.16 ± 5.81	7.00 ± 4.08	0.696	1.04 (0.84–1.28)
Women	61 (93.8%)	16 (94.1%)	0.999	0.95 (0.10–9.12)	45 (97.8%)	13 (100.0%)	0.999	1.28 (1.12–1.48)	16 (84.2%)	3 (75.0%)	0.999	1.77 (0.13–23.39)
Joint contractures	24 (36.9%)	4 (23.5%)	0.395	1.90 (0.55–6.50)	14 (30.4%)	2 (15.4%)	0.481	2.40 (0.47–12.30)	10 (52.6%)	2 (50.0%)	0.999	1.11 (0.12–9.60)
Raynaud’s phenomenon	63 (96.9%)	17 (100.0%)	0.999	-	44 (95.7%)	13 (100.0%)	0.999	1.29 (1.12–1.49)	19 (100.0%)	4 (100.0%)	-	-
Digital ulcerations	25 (38.5%)	2 (11.8%)	0.044	4.68 (1.002–22.25)	16 (34.8%)	1 (7.7%)	0.084	6.40 (0.76–53.76)	9 (47.4%)	1 (25.0%)	0.604	2.70 (0.23–30.84)
HAQ > 1.5	34 (52.3%)	2 (11.8%)	0.003	8.22 (1.74–38.89)	21 (45.7%)	1 (7.7%)	0.020	10.08 (1.20–84.05)	13 (68.4%)	1 (25.0%)	0.260	6.50 (0.55–76.17)
Muscle weakness	4 (6.2%)	1 (5.9%)	0.999	1.04 (0.11–10.04)	2 (4.3%)	1 (7.7%)	0.533	0.54 (0.04–6.53)	2 (10.5%)	-	0.999	-
Lung fibrosis	35 (53.8%)	7 (41.2%)	0.420	1.66 (0.56–4.91)	22 (47.8%)	5 (38.5%)	0.754	1.46 (0.41–5.16)	13 (68.4%)	2 (50.0%)	0.589	2.16 (0.24–19.27)
Increased systolic lung arterial pressure	16 (24.6%)	1 (5.9%)	0.107	5.22 (0.64–42.56)	11 (23.9%)	-	0.100	-	5 (26.3%)	1 (25.0%)	0.999	1.07 (0.08–12.83)
Renal crisis	8 (12.3%)	-	0.195	-	6 (13%)	-	0.322	-	2 (10.5%)	-	0.999	-
Positive antinuclear antibodies	50 (78.1%)	13 (76.5%)	0.999	1.09 (0.30–3.90)	33 (73.3%)	10 (76.9%)	0.999	0.82 (0.19–3.51)	17 (89.5%)	3 (75.0%)	0.453	2.83 (0.19–41.99)
Positive antitopoisomerase-1 antibody	13 (20.0%)	1 (5.9%)	0.280	4.00 (0.48–32.98)	3 (6.5%)	-	0.999	-	10 (52.6%)	1 (25.0%)	0.590	3.33 (0.29–38.08)
Positive anticentromeric antibodies	16 (25.8%)	7 (43.8%)	0.219	0.44 (0.14–1.39)	14 (31.8%)	6 (50.0%)	0.313	0.46 (0.12–1.70)	2 (11.1%)	1 (25.0%)	0.470	0.37 (0.02–5.57)
Elevated value of creatine phosphokinase (CPK)	3 (4.6%)	-	0.999	-	2 (4.3%)	-	0.999	-	1 (5.3%)	-	0.999	-
Increased acute phase reactants	25 (38.5%)	5 (29.4%)	0.580	1.50 (0.47–4.76)	15 (32.6%)	4 (30.8%)	0.999	1.08 (0.28–4.11)	10 (52.6%)	1 (25.0%)	0.590	3.33 (0.29–38.08)
Proteinuria	-	-	0.999	-	-	-	0.999	-	-	-	0.999	-

## Data Availability

The data presented in this study are available on request from the corresponding author.

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
