# Peer review of "Low-Frequency Magnetic Resonance Imaging Identifies Hand Joint Subclinical Inflammation in Systemic Sclerosis"

_diagnostics, 2022, doi:10.3390/diagnostics12092165_

Round 1

Reviewer 1 Report

The thesis is based on an interesting joint observation with MRI.

These errors and possible imaging predictions are being researched in many developed Western countries. I feel like some things are missing.

Inclusion and exclusion criteria should be detailed. Have you had a severe autoimmune disease? Have you ever had a patient with osteoporosis?

A pivotal question.

I really like the analysis and representation. The analysis defines the results well.

I miss the specification of the limitation separately from the thesis. Please revise it.

Nice job overall, congratulations.

Reviewer 2 Report

Any work on SS pathogenesis is significant, as we still lack a proper understanding of the disease. This paper, however, is a sidestep in that progress. Seeing RA as a comparator is puzzling – I would have expected healthy controls. Alternatively, seeing SS stages being assessed or followed up might be beneficial, so this somewhat deviates from the expectation. What about SS (sub)types? Always add a P= next to the numbers (…patients and swollen joints is missing P=). Minor linguistic edits are needed; the sentence structure is more Slavic than English, so the manuscript would benefit from more editing for easier reading. Why would you limit the sample to patients who reported hand pain? This will skew the results in a way that you will only have those who already have some clinical comlaint; it would have been much better to have all the SS in the sample and then explore the available data (also, favouring MRI over the clinical status is questionable, and in this combination, it is confusing to understand what approach you had in mind when targeting the sample and to who is this then generalizable)? Paragraphs in the Discussion section seem erratic; the authors should consider improving the text flow. Of note, we know about the joint affection in SS, so really, the purpose of the study seems even less clear to me now; I expected to see some multivariate analysis (regression), disease stage association/progression dynamics, control of the treatment applied in the analysis, or any of the combination of these ideas. As it stands now, the idea is limited, and the manuscript, although useful to some degree, does not manage to push the field forward. Therefore, I would kindly ask the authors if any of these ideas are applicable and if the data is available to provide a quantum step forward over what is known. We direly need any input on the SS pathogenesis, and somehow this entire dataset feels underexplored for what it could provide.

Round 2

Reviewer 2 Report

All is improved, but I would kindly ask the authors to change the following sentence: 

An estimation error level of less than 5% (p <0.05) was used as the threshold of statistical significance

into this: 

Significance was set at P<0.05

There is no need to confuse the readers with error levels, you can refer to is as the significance level. The rest is very good, and I commend the authors for performing additional analysis. 

Author Response

Dear Mr/Mrs,

Thank you for suggestion. I changed the text accordingly.

Kind regards